# Impact of Thermo-Responsive N-Acetylcysteine Hydrogel on Dermal Wound Healing and Oral Ulcer Regeneration

**DOI:** 10.3390/ijms25094835

**Published:** 2024-04-29

**Authors:** Andrew Padalhin, Celine Abueva, Hyun Seok Ryu, Seung Hyeon Yoo, Hwee Hyon Seo, So Young Park, Phil-Sang Chung, Seung Hoon Woo

**Affiliations:** 1Beckman Laser Institute Korea, College of Medicine, Dankook University, Cheonan 31116, Republic of Korea; drdrew200708@dankook.ac.kr (A.P.); cgabueva@dankook.ac.kr (C.A.); ryuhs2023@dankook.ac.kr (H.S.R.); ryuhs@dankook.ac.kr (S.Y.P.); pschung@dankook.ac.kr (P.-S.C.); 2Medical Laser Research Center, Dankook University, Cheonan 31116, Republic of Korea; 3School of Medical Lasers, Dankook University, Cheonan 31116, Republic of Korea; yoosh6653@dankook.ac.kr (S.H.Y.); shh8881@dankook.ac.kr (H.H.S.); 4Department of Otorhinolaryngology-Head and Neck Surgery, College of Medicine, Dankook University, Cheonan 31116, Republic of Korea

**Keywords:** hydrogel, n-acetylcysteine, oral ulcer, wound healing, inflammation

## Abstract

This study investigates the efficacy of a thermo-responsive N-acetylcysteine (NAC) hydrogel on wound healing and oral ulcer recovery. Formulated by combining NAC with methylcellulose, the hydrogel’s properties were assessed for temperature-induced gelation and cell viability using human fibroblast cells. In vivo experiments on Sprague Dawley rats compared the hydrogel’s effects against saline, NAC solution, and a commercial NAC product. Results show that a 5% NAC and 1% methylcellulose solution exhibited optimal outcomes. While modest improvements in wound healing were observed, significant enhancements were noted in oral ulcer recovery, with histological analyses indicating fully regenerated mucosal tissue. The study concludes that modifying viscosity enhances NAC retention, facilitating tissue regeneration. These findings support previous research on the beneficial effects of antioxidant application on damaged tissues, suggesting the potential of NAC hydrogels in improving wound care and oral ulcer treatment.

## 1. Introduction

Chronic wounds form when the usual sequence of the healing processes is disrupted. Individuals with preexisting medical conditions are especially prone to developing chronic wounds. Abnormalities in the healing of wounds impose significant physical and psychological burdens on a considerable portion of the population, including the elderly, individuals with diabetes, and patients undergoing immunosuppressive treatments, chemotherapy, or radiotherapy for cancer [1]. One major factor that plays a role in the functional regeneration of damaged tissues is oxidative stress. Reactive oxygen species (ROS) have been widely acknowledged for their dual role in the wound-healing process, acting both as harmful and beneficial agents [2,3]. Additionally, continuous oxidative stress is linked to alterations in proteins and lipid peroxidation, which have been demonstrated to contribute to sustained low-level inflammation hindering the natural progression of wound healing and leading to elevated cell apoptosis and senescence [4,5,6].

Free radicals represent highly volatile molecules, with ROS comprising a subset containing oxygen atoms alongside reactive entities like superoxides and peroxides [7]. While typically generated as metabolic byproducts and as responses to pathogens, an excess production triggers an augmented burden of free radicals and ROS, culminating in oxidative stress. Although ROS generation is a physiological process, excessive output can prove detrimental [7,8]. Moderate levels of ROS are necessary for an effective defense against pathogens and for cellular signaling such as angiogenesis. However, excessive ROS production or a compromised antioxidant defense system lead to oxidative stress, a key factor contributing to impaired wound healing [7]. The body’s physiological defense mechanisms against oxidative stress encompass ROS-neutralizing enzymes like superoxide dismutase, catalase, glutathione peroxidases, and peroxiredoxins. Additionally, both endogenous and exogenous low-molecular-weight antioxidants, such as glutathione, vitamin E, vitamin C, and phenolics, serve as non-enzymatic safeguards against ROS [5,9]. These antioxidants willingly undergo oxidation, transforming into radicals themselves, albeit less reactive and hence less damaging than the radicals they scavenge. Operating in a continuous cycle, antioxidants collaborate to regenerate the sacrificed counterparts, ensuring their availability in their reduced forms for cellular defense mechanisms [10].

Diminished levels of these antioxidants have been linked to impaired wound healing [2,3]. Consequently, antioxidants, particularly those derived from natural sources, are theorized to mitigate oxidative stress in wounds, and thereby facilitate the healing process. Antioxidants play a crucial role in mitigating potential damage inflicted upon biological molecules such as DNA, proteins, lipids, and bodily tissues in the presence of reactive species [3]. Their effectiveness lies in their relatively unreactive nature compared to free radicals, thus diminishing the likelihood of propagating harmful effects.

The regulation of redox balance, achieved by adjusting levels of reactive oxygen species (ROS) and antioxidants, is a focus of novel therapeutic approaches. Enhancing wound healing may be facilitated by antioxidant agents that sustain safe levels of ROS within the wound environment [2,3,11,12,13,14]. Consequently, there is a burgeoning interest in employing antioxidant compounds for wound management, leading to the development and evaluation of numerous biomaterials [13,15,16].

N-acetylcysteine (NAC) is a drug approved by both the FDA and the WHO that is essential for treating acetaminophen overdose and, more recently, has been used as a mucolytic agent in respiratory diseases [17,18,19]. As a nutritional supplement, NAC is widely available without a prescription in several countries due to its antioxidant properties and commercial appeal as a nutraceutical [17]. Its primary function lies in its antioxidant and anti-inflammatory capabilities, crucial for maintaining cellular redox balance and providing therapeutic potential for diseases associated with oxidative stress [1,13,20,21]. Previous research has demonstrated NAC’s efficacy in protecting against oxidative stress and inflammation across various conditions. These include mitigating brain damage induced by transient cerebral ischemia [6], managing pain and inflammation during infections [22], and restoring thyroid morphology by reducing inflammatory cell infiltration [23].

This study focuses on the development of a topical thermo-responsive NAC hydrogel and evaluates its efficacy for improving tissue regeneration in dermal wounds and oral ulcers. This investigation covers the optimization of the NAC hydrogel formulation in terms of its desired gelling property to enhance retention upon application on damaged tissues, primarily centered on in vitro and in vivo biocompatibility.

## 2. Results and Discussion

### 2.1. Initial Testing Using Commercially Available NAC Solution

Before formulating NAC hydrogels by combining N-acetylcysteine (NAC) with methylcellulose (MC), in vitro cell viability was tested using different dilutions of commercially available NAC solution (Mucomyst). In vitro testing with the commercial NAC solution indicated that beneficial effects on viability are observed from 1–10% NAC when tested with fibroblast cells. While no significant difference among 1–10% NAC was observed, viability steeply dropped when fibroblasts were exposed to the highest concentration (20%) of NAC (Figure 1A).

Based on the context of these results, NAC emerges as a double-edged sword in its interaction with oxidative stress and cellular viability. Few studies have indicated some negative effects of using NAC. In previous studies involving chronic myeloid leukemia (CML), NAC’s potential as a ROS scavenger enhanced the efficacy of drugs like imatinib [1,20,24,25]. Although it indeed intensified imatinib-induced cell death, the mechanism behind this enhancement proved different than initially hypothesized, primarily involving increased expression of endothelial nitric oxide synthase rather than direct ROS scavenging. In another study about idiopathic pulmonary fibrosis (IPF), NAC’s role in reducing oxidative stress was observed to decrease epithelial cell viability, raising concerns about its standalone usage [26]. In said study, the combination of NAC with curcumin, despite its potential as an antioxidant and antifibrotic agent, showed no significant improvement in cell viability, further complicating its application. This was even exemplified in studies exploring the NAC’s capacity to protect oocytes [27] and endothelial cells [28]. In the former study, NAC has been shown to be beneficial when used at 1.0 mM concentration but can significantly affect cellular ultrastructure and reduce the viability of bovine secondary follicles when used at concentrations greater than 5.0 mM [27]. In the latter study about endothelial cells, although low concentrations of NAC have been shown to offer protection against oxidative stress induced by proinflammatory cytokines, high concentrations trigger a biphasic response, leading to increased ROS production, protein carbonylation, and glutathionylation, highlighting the complex interplay between antioxidants and cellular responses [28].

These findings underscore the need for a nuanced approach to utilizing NAC and other antioxidants, especially concerning concentration levels, to mitigate potential adverse effects and optimize therapeutic outcomes. The results of the in vitro test indicate similar effects with increasing concentrations of commercially available NAC formulation, demonstrating the biphasic effect of NAC on fibroblast cells and echoing the need for optimizing NAC concentration for different use cases.

Initial testing was conducted by combining commercial grade 20% NAC solution (Mucomyst) with methylcellulose (2% weight/volume) solution at varying volume compositions. Figure 1B shows the different volume compositions of NAC hydrogel before and after gelation at 37 °C. Our initial observations indicate that combining NAC and the commercial MC at 50–50% and 75–25% volume ratios are suitable for creating solutions that can form a stable gel after incubation at 37 °C. However, increasing concentrations up to 10% do not provide additional benefits in terms of fibroblast viability. Rather, further increasing the concentration of NAC in this formulation shows a downward trend in terms of fibroblast viability.

Focusing on the aspect of increasing retention of antioxidants on the surface of wounds and oral ulcers, different formulations of thermo-gelling NAC hydrogels were created. This was accomplished by combining different volume percentages of NAC and methylcellulose. Methylcellulose (MC) is a remarkable biomaterial in the medical field, owing to its distinctive thermo-responsive characteristics and versatile applications [29,30,31,32,33]. Its synthesis pathways result in either a homogeneous or heterogeneous distribution of methoxy groups along the macromolecule, influencing its properties and potential uses [29,34]. MC hydrogels, prepared from aqueous solutions, exhibit a reversible sol–gel transition in response to temperature variations, rendering them valuable for medical applications where controlled release or modulation of properties is desired [29,30]. In the realm of drug delivery, MC finds use as a viscous ultrasonic coupling medium for transdermal sonophoresis, aiding in the delivery of insulin and vasopressin [32]. Its application extends to tissue engineering, where it serves as a building block for scaffolds in injured brain tissues [33], thanks to its ability to form hydrogels with tailored mechanical properties. Moreover, in ophthalmology, MC assists in reducing drug solution drainage from the eyes of albino rabbits, showcasing its potential in ocular therapies [35]. MC hydrogels also demonstrate biocompatibility, as evidenced by in vitro tests with fibroblast and mesenchymal stem cell lines, indicating their non-cytotoxic nature and suitability for tissue engineering applications [36]. The thermo-responsive nature of MC hydrogels further enhances their appeal, offering stimuli-responsive functionalities for designing innovative scaffold/cell delivery systems. The crosslinking mechanisms of MC involve the formation of hydrophobic interactions, leading to the creation of a three-dimensional network that can be tailored by adjusting parameters such as concentration and temperature [31,36]. MC’s stability in physiological environments and non-toxicity to cells make it an attractive candidate for medical use. Its mechanical properties can be easily adjusted to match those of native tissues, enhancing its compatibility and efficacy in various biomedical applications [29,30,34,36].

### 2.2. Optimization of NAC–MC Hydrogel Formulation

After confirming the suitable range of MC and commercial NAC volume ratio to create a thermo-gelling material, we modified the formulation by using pure NAC powder in place of the commercial solution. In addition, we also adjusted and tested the methylcellulose concentration down to 1% weight/volume to further minimize said component. Figure 2 shows the different formulations of NAC hydrogels before and after gelation. All formulations were tested at 50:50% volume solutions of NAC and MC. Taking the results of the in vitro test conducted with the commercial NAC solution, we opted to utilize lower concentrations of NAC for the resulting hydrogel to avoid the biphasic effect of NAC at higher concentrations [24,26,27,28]. Results of the gelation test indicated that all formulations were capable of forming a stable gel with increased temperature.

In order to comprehensively validate the efficacy and applicability of the diverse NAC hydrogel formulations, an extensive and meticulous evaluation was undertaken by subjecting extracts of each formulation to human fibroblast cells, as elucidated in Figure 3A. The findings of this exhaustive investigation yielded crucial insights, revealing that the exposure of fibroblast cells to concentrations of 1% and 2% methylcellulose exerted a negligible impact on fibroblast viability. However, as the concentration of NAC within the hydrogel formulations escalated, a discernible trend emerged, reminiscent of that observed with the commercial NAC solution in hydrogels containing 1% MC. Particularly noteworthy was the observation that solely MC2-NAC1 exhibited the highest viability among the hydrogels containing 2% methylcellulose. Moreover, a significant enhancement in viability was distinctly discerned in cells exposed to MC1-NAC1, MC1-NAC2.5, and MC1-NAC5 formulations. Intriguingly, across NAC hydrogels featuring NAC concentrations ranging from 1% to 5%, no substantial variance in cell viability was observed. These compelling findings serve to underscore the pivotal role of NAC hydrogel compositions comprising 1% MC and at least 1% NAC in bolstering fibroblast viability. Furthermore, the demonstrable efficacy of these formulations was vividly illustrated through the meticulous examination of fluorescently stained images of cells cultured with NAC hydrogel containing 1% methylcellulose, as depicted in Figure 3B. Notably, a discernible reduction in the number of cells was observed in wells exposed to 20% Mucomyst compared to those subjected to lower concentrations (ranging from 10% to 1%). Additionally, a substantial decrease in attached cells was noted in wells exposed to NAC hydrogel containing 1% methylcellulose and 10% NAC. However, cells exposed to lower concentrations of NAC (ranging from 5% to 1%) exhibited a considerable number of cells in comparison to wells exposed solely to methylcellulose or the control group, further accentuating the nuanced effects of varying NAC concentrations on cell viability.

### 2.3. Effect of NAC Hydrogel on Fibroblast ROS

The antioxidant effect of the NAC hydrogel was tested against the commercial product and NAC solution at 5% concentration. Chemically induced intracellular ROS was measured in fibroblast cells exposed to hydrogen peroxide. Figure 4A,B show the images of fibroblast cells treated with H2DCFDA after exposure to different preparations of NAC. Results indicate that all NAC treatments were relatively effective in reducing chemically induced intracellular ROS in fibroblast cells. Both NAC-5 and MC1-NAC5 treatments resulted in lower ROS-stimulated fluorescence compared to the commercial product. However, statistical analyses of the relative mean fluorescence (Figure 4C) from the sampled images revealed no significant difference among the treatment groups. These results indicate that even with the modified preparation of NAC in the form of the NAC hydrogel, its capacity to reduce intracellular ROS is not affected, and the addition of the methylcellulose does not affect its overall biocompatibility. In addition, the lower concentration of N-acetylcysteine in the NAC hydrogel is much more beneficial than the higher concentrations found in the commercial product.

### 2.4. Effect of NAC Hydrogel on Dermal Wound

Macroscopic images depicting the full-thickness wounds were meticulously analyzed to discern the impact of various topical treatments, as depicted in Figure 5A. In this experimental setup, the routine application of saline solution served as the positive control for comparison purposes. Furthermore, Mucomyst 20% (MUC-20) and 5% N-acetylcysteine (NAC-5) were utilized as additional comparative agents alongside the NAC hydrogel comprising 1% methylcellulose and 5% NAC (MC1-NAC5). A comprehensive evaluation of the wound beds upon gross examination initially yielded no discernible distinctions among the different treatment modalities. However, noteworthy disparities emerged on the final day of observation, particularly evident in the wound treated with MC1-NAC5, which exhibited a marked reduction in dimensions. This notable observation was further substantiated through meticulous image analyses of the macroscopic images. Notably, the normalized wound area measurements depicted in Figure 5B underscored the significant efficacy of routine MC1-NAC5 application, manifesting an approximate 85% reduction in the overall wound area. In contrast, other treatment groups only managed to curtail wound size by approximately 75%. Nevertheless, it is imperative to acknowledge that this substantial reduction was discerned solely towards the culmination of the 1-week observation period.

H&E-stained tissue sections of the wound bed from the different treatment groups show substantial differences in tissue development (Figure 6). Notably, the distance between the wound margins of the saline group and the MUC-20 was more pronounced than that of tissue samples from NAC-5- and MC1-NAC5-treated groups. Thin fibrous tissue can also be observed in the dermal tissue layer of the saline- and MUC-20-treated tissue, while NAC5- and MC1-NAC5-treated wounds show the development of more dense fibrous tissue. Wounds treated with the different N-acetylcysteine solutions (MUC-20/NAC5/MC1-NAC5) resulted in thicker epidermal tissue formation, with a noticeably developed thin outer corneal layer compared to the saline-treated group. MC1-NAC5 resulted in substantially thicker epidermal tissue formation within the central wound surface (red box) and along the wound margin (blue box).

### 2.5. Effect of NAC Hydrogel on Oral Ulcer

The effect of routine application of NAC hydrogel on oral ulcers was thoroughly evaluated utilizing an animal model of chemically-induced oral ulcers. Throughout the 1-week observation period, the size and overall condition of the oral ulcers were meticulously assessed, with photographs being captured for documentation purposes. The findings from the oral ulcer size evaluation (as depicted in Figure 7A) elucidate a noteworthy reduction in ulcer size with the consistent utilization of MC1-NAC5 in contrast to all other experimental groups. Interestingly, the application of an equivalent concentration of NAC (5%) demonstrated a marginal improvement in oral ulcers; however, it failed to exhibit statistical significance compared to the control group. Conversely, the routine application of 20% Mucomyst yielded highly variable outcomes, showcasing minimal deviation from the control group (as illustrated in Figure 7B).

Examination of the histological preparation of the extracted oral mucosal tissue provides deeper insights into the discernible variances in tissue regeneration observed among the diverse treatment regimens. Notably, all samples exhibited considerable fibrous tissue formation within the papillary layer. Interestingly, despite the routine application of saline, MUC-20, and NAC-5, there was no notable reduction in the mucosal layer gap. However, it is noteworthy that treatment with NAC-5 elicited a distinctive thicker fibrous tissue morphology compared to the other treatments. Particularly compelling was the outcome observed with routine application of MC1-NAC5, wherein near full closure of the oral ulcer was achieved, accompanied by a regenerated mucosal layer of comparable thickness (as illustrated in Figure 8). This finding underscores the remarkable potential of MC1-NAC5 in promoting comprehensive tissue regeneration within the oral mucosa, highlighting its promising therapeutic efficacy in the management of oral ulcers.

### 2.6. Effect of NAC Hydrogel on Oral Ulcer Inflammation

Considering that the NAC hydrogel showed a pronounced effect as an oral ulcer treatment, in addition to observing the resulting tissue microarchitecture from the different treatment groups, tissue expression of pro-inflammatory cytokines and tissue remodeling were also examined through immunostaining (Figure 9). MMP-2 was used as a maker for active tissue remodeling [37,38,39] while IL-6 and TNF-α were observed as pro-inflammatory markers [40,41,42,43,44,45]. IHC staining revealed elevated expression of MMP-2 along the fibrous tissue formation in all treatment groups, particularly in tissues treated with NAC-5 and MC1-NAC5. However, neither was significantly higher than the other. Although oral ulcer tissues treated with the commercial NAC solution showed higher expression of MMP-2, it was not significantly different from that of the control group. Examination of IL-6 and TNF-α through immunofluorescence staining showed drastically different outcomes across all treatment groups. Oral ulcers treated with the MUC20 showed significantly lower expression of IL-6 with MC1-NAC5. NAC-5 treatment resulted in relatively higher expression of IL-6 compared to all treatment groups and the control group. Examination of TNF-α revealed a reversed situation between MUC20 and NAC-5 groups, with MUC 20 showing the highest expression of TNF-α among all treatment groups whilst not significantly differing from the control. Expressions of both aforementioned pro-inflammatory markers in MC1-NAC5-treated oral ulcers were significantly lower among all groups. These results demonstrated the importance of optimal exposure of damaged tissues to antioxidants to improve tissue regeneration and address chronic inflammation.

The results of the in vivo experiments on wound healing and oral ulcers exhibit the capacity of NAC hydrogel for improving tissue regeneration. Viscosity and concentration optimization are considered crucial to the realization of NAC as a topical treatment for damaged dermal and mucosal tissue. The significance of viscosity in drug formulations cannot be overstated, particularly in enhancing drug contact time, and thereby improving therapeutic outcomes. Several studies have shown the pivotal role of viscosity in topical, ocular, and buccal drug delivery systems [46,47,48,49,50].

Starting with topical treatments, formulations for conditions like psoriasis often struggle with low viscosity, leading to poor retention on the skin surface and consequently low drug delivery efficiency. To address this, one study investigated the development of a water-responsive gel that exhibits a unique property of transitioning from a liquid to a gel state upon contact with water, thereby increasing its viscosity [46,50]. This enhances skin retention and drug permeation, making it highly suitable for topical drug delivery. Similarly, in ocular drug delivery, where bioavailability can be a challenge, viscosity plays a crucial role in prolonging drug residence time. Liquid and semisolid formulations are preferred over solid ones due to their ability to incorporate viscosity-enhancing agents and mucoadhesive excipients [48]. Cellulose derivatives, particularly hydroxy methylcellulose, have been extensively utilized in ophthalmic preparations for their thickening properties and compatibility with drugs, leading to increased contact time and improved therapeutic outcomes. Equally important is buccal drug delivery in which limited absorption area and mucosal barrier properties pose challenges. Viscous liquids serve as effective protectants or drug delivery vehicles to coat the buccal surface, enhancing drug absorption over time [49,50,51,52]. Mucoadhesive polymers play a crucial role in maintaining prolonged contact between the formulation and the oral mucosa, thereby counteracting the saliva wash-out effect and improving bioavailability. The choice of mucoadhesive agents, such as poly(acrylic acid) derivatives [53] and cellulose derivatives, is critical to ensure biocompatibility and non-toxicity [54,55]. Excipients should be carefully selected to ensure the solubility and stability of liquid formulations, while newer formulations like oral thin films and wafers are being explored to further improve contact with the oral mucosa [54]. As demonstrated in the in vitro cell viability and ROS assay, combining of the methylcellulose with the N-acetylcysteine to create the thermo-gelling NAC hydrogel did not significantly affect its biocompatibility and capacity to reduce intracellular ROS. Hence, the methylcellulose component of the NAC hydrogel serves more of a viscosity/physical form modifier with limited chemical interaction with the N-acetylcysteine component.

Overall, the role of viscosity in drug formulations is multifaceted, influencing drug release, retention, absorption, and ultimately therapeutic efficacy. By understanding and harnessing the properties of viscosity, researchers continue to innovate in drug delivery systems, aiming to enhance patient outcomes across various medical conditions. These effects were particularly observed from the application of NAC hydrogel on animal models of chemically-induced oral ulcers. A significant reduction in ulcer size was observed after routine treatment with NAC hydrogel [51,56,57] compared to higher concentrations in the commercial formulation or the non-modified NAC solution with the same concentration. The reduction of pro-inflammatory tissue markers (IL-6 and TNF-a) and increase in the tissue remodeling marker (MMP-2) observed from the IHC and immunofluorescence staining indicate that this type of topical NAC preparation can indeed provide additional support during the healing phase. A literature review of previous studies indicates that NAC treatment generally results in reduction of MMP-2 in patients with Crohn’s Disease [58], diabetic rats [59], and mice with atherosclerotic plaque formations [60]. However, our current results are contradictory to these initial findings. It should be noted that most of these studies were conducted in the context of inflammatory diseases known to have excessive expression of MMP-2. Despite the fact that MMP-2 is highly correlated to abnormal tissue growth and cancer metastasis, like other enzymes, MMP-2 has also been found to contribute in in hastening wound healing particularly by increasing cell migration [39,45].

Equally, previous studies have already noted that NAC treatment is capable of significantly reducing IL-6 expression in patients with ischemic stroke [61], hemorrhagic shock [62], and rheumatoid arthritis [63]. Likewise, NAC treatment have also been found to reduce expression of TNF-α in patients with sarcoidosis [64] and in COVID-19 patients [65]. The reductions in MMP-2, IL-6, and TNF-α described in these studies are all anchored in the central mechanism of NAC’s anti-oxidative activity leading to anti-inflammatory effects. NAC can act as an antioxidant in two ways. As a thiol it is an ideal precursor of l-cysteine and reduced glutathione (GSH). NAC is readily converted to cysteine upon absorption in the cells, which increasing the supply of precursor for the synthesis of intracellular GSH. Thus, it can directly act to replenish the intracellular reduced GSH, which is exhausted under high oxidative stress and inflammation [66,67,68]. Alternatively, NAC can serve as a direct scavenger of free radicals such as hydrogen peroxide, superoxides, and hydroxyl radicals [69,70]. By replenishing the cellular redox balance, NAC is capable of influencing the function of redox-sensitive cell-signaling and transcription pathways like Nuclear Factor–kB (NF-kB), which controls numerous pro-inflammatory genes such as IL-6, IL-8, and TNF-α, which in turn regulate the production of immune cells and inflammatory T cells [71,72].

While the current data have demonstrated the effectiveness of NAC hydrogel in facilitating enhanced dermal and oral ulcer regeneration, this paper does not delve into the complete mechanism underlying its impact on tissue development. Like any other bioactive compound, there are also specific conditions under which NAC-based treatments might not be fully applicable. Previous studies have noted the anti-coagulant effect of NAC [73,74], which could be a major limiting factor when considering the application of NAC hydrogel. While it is a concern worth looking into, the concentration of NAC used in the current study is way below the concentration needed to severely compromise blood clotting in open wounds, as exemplified by the in vivo results. This type of treatment would ideally be limited to post-operative management of wounds. Application of NAC hydrogel during surgery remains to be determined, pending further testing. The same can be said for potential drug interactions, particularly to certain medications that also alter glutathione levels. Furthermore, the long-term effects of NAC hydrogel application on wound healing and oral ulcer treatment remain unexplored, leaving the potential influence of this antioxidant on functional tissue remodeling unexplored. Thus, future research endeavors are warranted to investigate various formulations of NAC hydrogel. A summary of findings is listed in Table 1.

## 3. Materials and Methods

### 3.1. Formulation of Thermo-Gelling NAC Hydrogel

A common high-concentration NAC solution, Mucomyst 20% (MUC-20) (Boryung Biopharma, Seoul, Republic of Korea), was purchased commercially. Fibroblast cells were used to test the in vitro biocompatibility of the thermo-responsive NAC hydrogel formulations. Briefly, human fibroblast cells (CRL-2522) (ATCC, Manassas, VA, USA) were cultured in Eagle’s Minimum Essential Media (EMEM) (Corning Inc., New York, NY, USA) supplemented with 1% penicillin-streptomycin (Corning Inc., New York, NY, USA) and 10% fetal bovine serum (Corning Inc., New York, NY, USA). After the cells reached confluence, the fibroblast cells were then seeded into a 96-well plate at a density of 10,000 cells/well 24 h before viability testing. Then, 200 µL of the test media containing a 10% volume ratio of Mucomyst diluted at 10%, 5%, 2.5%, and 1% was added in each well (*n* = 6). The cells were incubated with the test media for another 24 h, after which cell viability testing was conducted using an EZ-Cytox Assay kit (DoGenBio, Seoul, Republic of Korea). Briefly, the test media were removed from each well of the 96-well plate then 200 µL of EMEM containing 10% by volume of CCK8 solution was added in each well. The plate was then incubated for an additional 2 h to develop the formazan dye from the viable cells. Using a microplate reader (TECAN), absorption readings from each were taken at 450 nm.

Methylcellulose (M0512-100G) and N-acetylcysteine (A7250-100G) were procured from Sigma Aldrich, St. Louis, MO, USA. Methylcellulose solution, 1% and 2% weight by volume, was prepared by adding 2.0 mg and 4.0 mg MC powder, respectively, to separate tubes containing 40 mL of autoclaved distilled water. The solution was thoroughly mixed using a tube inverter overnight. The NAC solution was prepared by dissolving 20% weight by volume of NAC powder in autoclaved distilled water, followed by heating in a 60 °C oven for 10 min until all solutes were fully dissolved. The resulting highly acidic NAC solution was then pH adjusted to 7.0 using 10 M hydrochloric acid (Daejung Chemicals, Siheung, Republic of Korea). After pH adjustment, the MC solutions (1% or 2%) were combined with the NAC solution at a 50:50 volume-by-volume ratio. NAC concentration was adjusted based on the total volume of the combined MC and NAC. To test the gel formation from the different formulations, 2 mL samples were placed in 5 mL snap tubes. The snap tubes were then placed in a 37 °C water bath for 5 min. The gelation of the samples was visualized through photographs of inverted snap tubes.

### 3.2. In Vitro Biocompatibility of NAC Hydrogel

Fibroblast cells were used to test the in vitro biocompatibility of the thermo-responsive NAC hydrogel formulations. Briefly, human fibroblast cells (CRL-2522) (ATCC, Manassas, VA, USA) were cultured in Eagle’s Minimum Essential Media (EMEM) (Corning Inc., New York, NY, USA) supplemented with 1% penicillin-streptomycin (Corning Inc., New York, NY, USA) and 10% fetal bovine serum (Corning Inc., New York, NY, USA). After the cells reached confluence, the fibroblast cells were then seeded into a 96-well plate at a density of 10,000 cells/well 24 h before viability testing. The test media containing the hydrogels were prepared by adding 1.0 mL of hydrogel sample in individual 15 mL tubes which were then incubated in a 37 °C water bath for 5 min. Upon gelling of the hydrogels, 9.0 mL of complete cell media were carefully added into each tube and was then returned to a 37 °C water bath for another 10 min. Cell media were then removed from the cells seeded in the 96-well plate and the cells were washed with PBS. In each well, 200.0 µL of test media was added and the cells were incubated for another 24 h. A cell viability assay was conducted (EZ-Cytox, DoGenBio, Seoul, Republic of Korea) following the previously stated manufacturer method. F-actin staining was performed using phalloidin 488 stain to visualize the cells after exposure to the different NAC hydrogels. The CCK8 solution was removed from the well plate and the cells were fixed with 4% formalin for 30 min. The formalin solution was then removed and the cells were washed with phosphate-buffered saline (PBS). The cells were then permeabilized with 0.5% Triton-X for 10 min followed by a PBS wash. Blocking was then performed using 3% BSA solution for 1 h, followed by another PBS wash. Phalloidin stain conjugated with Alexa Fluor 488 was then applied on the cells for 30 min, followed by DAPI (4′,6-diamidino-2-phenylindole), a blue fluorescent DNA stain for contrasting the nuclear component of the intact cells. Images of the fluorescently stained cells were taken using EVOS M7000 (Invitrogen, Waltham, MA, USA).

### 3.3. ROS Fluorescence Detection

A Reactive Oxygen Species Detection Assay Kit (H2DCFDA) (K936-250 BioVision Inc., Milpitas, CA, USA) was used to observe the effect of the different NAC formulations and NAC hydrogel on the intracellular ROS in human fibroblast cells (CRL-2522) (ATCC, Manassas, VA, USA) in vitro. Fibroblast cells were seeded in a 96-well plate, with each well containing approximately 2.5 × 10^4^ cells. Each group was designated with 6 wells. ROS kit staining was then performed according to the manufacturer’s protocol with relevant modifications. All cells were incubated with 100 µL of 1× ROS label for 45 min at 37 °C. The ROS label was then removed and the cells were exposed to 0.03% hydrogen peroxide (H_2_O_2_) with or without a 10% volume of either MUC20, NAC5, or MC1-NAC5 for 1 h. Wells treated with saline were used as a control. Micrographs of the fibroblast cells were then taken immediately after 1 h of incubation using an EVOS M7000 imaging system (Invitrogen, Waltham, MA, USA). Fluorescence intensity analysis of the images was performed by separating the image channels of the micrographs and analyzing mean pixel values (*n* = 6).

### 3.4. Animal Experiments

All animal experiments were conducted in accordance with the guidelines set by the Institutional Animal Care and Use Committee at Dankook University (DKU-23-082). A total of 36 male six-week-old Sprague Dawley rats weighing approximately 200–250 g were purchased from Orientbio Co., Songnam, Republic of Korea. The animals were allowed to acclimatize for 1 week before experimentation. Two rats were housed in each cage and placed in a climate-controlled animal room with a 12 h light–dark cycle. Food and water were provided as needed. The animals were then randomly distributed into 4 treatment groups between two experimental procedures.

#### 3.4.1. Induction and Treatment of Dermal Wound

Effect of NAC hydrogel on wound healing: The effect of the NAC hydrogel on skin tissue regeneration was observed using an animal model of full-thickness excision. Sixteen rats were randomly distributed into 4 different treatment groups (*n* = 4): Positive control (saline); Mucomyst 20% (MUC20); 5.0% N-acetylcysteine (NAC-5); and 5.0% N-acetylcysteine hydrogel with 1.0% methylcellulose (MC1-NAC5). Rats were anesthetized via inhalation of 2.0–3.5% isoflurane (Hana Pharm. Co., Ltd., Seoul, Republic of Korea) mixed with 100% oxygen. The rats’ backside side was cleared by shaving off the fur, followed by the application of hair removal cream (Trend Master, Hwaseong, Republic of Korea) for 10 min. The hair removal cream was then scraped off and an ethanol solution (70%) was used to clean the bare skin surface and clear any excess hair removal cream. Using the spine as a landmark, two identical full-thickness skin defects were created on both sides of each animal using a 10.0 mm biopsy punch (Kai Medical, Seki, Japan). Upon creation of the skin defect, 1.0 mL of test samples was applied on the surface of each wound for at least 3 min. After the application of the test samples, the animals were returned to their respective cages and allowed to fully recover. Excision wounds were photographed every other day before routine treatment with respective samples.

#### 3.4.2. Induction and Treatment of Oral Ulcer

Effect of NAC hydrogel on oral ulcer: An animal model of a chemically-induced oral ulcer was used to observe the impact of the thermos-responsive NAC hydrogel on oral mucosa regeneration. The remaining 16 rats were again divided into the 4 treatment groups previously mentioned in the wound healing experiment. The rats were first anesthetized using isoflurane inhalation. Upon complete anesthetic induction, the animal was laid on its right side under an operating microscope (Leica Biosystems, Wetzlar, Germany) with its posterior end situated away from the experimenter and its head towards the microscope field of view. With the animal’s oral cavity in view, the tongue was carefully retracted to the left side to expose the surface of the right inner cheek. An oral ulcer was then chemically induced by carefully injecting 15 µL of 60% acetic acid using a 300 µL tuberculin syringe into the mucosal tissue of the rat. The animal was then allowed to recover in its cage. The oral ulcer was allowed to develop over the course of 3 days. Application of the specified treatment was performed under anesthesia inhalation, during which 500 µL of saline (Control), Mucomyst 20% (MUC20), 5.0% N-acetylcysteine (NAC-5), or 5.0% N-acetylcysteine hydrogel with 1.0% methylcellulose (MC1-NAC5) was carefully pipetted into the oral cavity of each rat. After pipetting each sample into the oral cavity, the animal was then transferred into its home cage for recovery. Morphology and size of the oral ulcers were documented through photographs before routine application every other day for one week.

### 3.5. Histological Analyses

Tissue samples were extracted after 7 days, frozen, and embedded in optimal cutting temperature (OCT) medium (Sakura Finetek, Torrance, CA, USA) for sectioning. Frozen tissue samples were then cut into 5.0 µm sections using a Leica RM2135 microtome (Leica Biosystems, Wetzlar, Germany) and stained with hematoxylin and eosin (H&E). Images of the stained tissue sections were taken using an Olympus BX53 light microscope with cellSens imaging software (Standard version 2.3, Build 18987) (Olympus Life Science, Tokyo, Japan). Immunofluorescence (IF) and immunohistochemical (IHC) staining were also performed to visualize the expression of tissue repair marker MMP-2 and the pro-inflammatory markers IL-6 and TNF-α. Heat-induced antigen retrieval was performed on deparaffined and rehydrated tissue sections using citrate buffer (10 mM, pH 6.0). The tissue sections blocked with 3% bovine serum albumin (BSA) and individual slides were then incubated with either MMP-2 (ab86607, Abcam, Cambridge, UK), IL-6 (ab9324, Abcam, Cambridge, UK), or TNF-α (ab66579, Abcam, Cambridge, UK) primary antibodies at 4 °C overnight. Sections incubated in MMP-2 antibody were then treated with horseradish peroxidase-conjugated secondary antibody, followed by a color-developing reagent from the EnVision Detection System Peroxidase kit (Dako Co., Fort Collins, CO, USA). Hematoxylin counter was also performed before mounting and imaging using an Olympus BX53 light microscope with cellSens imaging software (Build 18987, Olympus Life Science, Tokyo, Japan). Fluorescence staining for sections stained with either IL-6 or TNF- α was performed using Alexa Fluor 555 (A21422/A241428, Invitrogen, Waltham, MA, USA) with DAPI for nuclear counterstaining. Micrographs of the immunofluorescence-stained tissue sections were then taken using the EVOS M7000 imaging system (Invitrogen, Waltham, MA, USA). Images were imported into FIJI software (ImageJ, version 1.54f) for histomorphometric analysis. Images from each test group were used to compare the fibrous tissue and epidermal tissue thickness. IHC-stained sections were evaluated using H DAB color deconvolution, followed by thresholding and area fraction measurement (*n* = 4). Fluorescence intensity analysis of IF-stained sections was performed by separating the image channels of the micrographs and analyzing mean pixel values (*n* = 4). Measurements were tabulated and statistically compared among the different test groups.

### 3.6. Statistical Analyses

Mean values along with their respective standard deviations were utilized to express measurement and scoring data. The software GraphPad Prism version 8.4.3 for Windows (GraphPad Software, San Diego, CA, USA) was employed for both plotting and analyzing data points. Statistical analysis involved conducting one-way analysis of variance (ANOVA) tests, followed by the Tukey test for multiple comparisons among groups. A significance threshold of *p* ≤ 0.05 was applied to determine statistical significance in all analyses.

## 4. Conclusions

This study focused on the development of a thermo-responsive NAC hydrogel formulation and its efficacy in promoting tissue regeneration in dermal wounds and oral ulcers. The initial tests demonstrated the potential of NAC hydrogel in enhancing fibroblast viability, reduction of intracellular ROS, and improving wound size reduction in animal models. Histological analyses reveal improvements in tissue development, increased expression of a tissue remodeling marker and reduction of pro-inflammatory markers with NAC hydrogel treatment compared to solutions containing high and low NAC concentrations. This study also further supports the importance of choosing the suitable concentration of excipient for viscosity modification of NAC. While the study underscores the therapeutic potential of NAC hydrogel, further investigation into its mechanism of action and long-term effects on tissue remodeling is warranted to optimize its clinical applications.

## Figures and Tables

**Figure 1 ijms-25-04835-f001:**
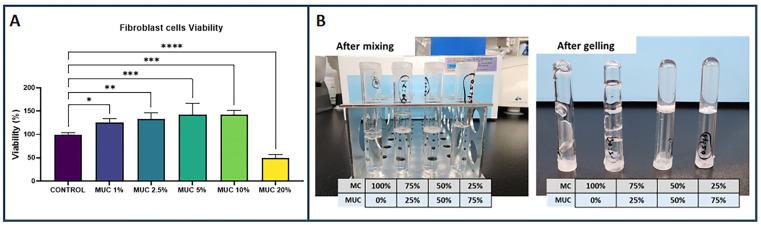
Viability test conducted with human fibroblast cells indicate that exposure to 1–10% NAC is beneficial but 20% NAC from commercially available solution does not offer further benefits (**A**) (*n* = 5, * *p* < 0.05, ** *p* < 0.01, *** *p* < 0.001, **** *p* < 0.0001). Commercially available solutions of NAC were mixed with 2% methylcellulose at different volumetric ratios to determine the suitable composition that would form a stable gel after incubation at 37 °C (**B**).

**Figure 2 ijms-25-04835-f002:**
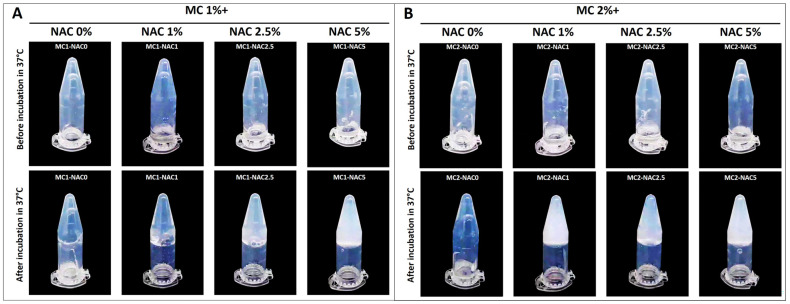
Different compositions of NAC hydrogel tested with lowered concentrations of pure NAC. Results indicate that NAC can be combined with 1% (**A**) to 2% (**B**) MC at a 50% volume ratio to create an NAC hydrogel capable of gelling at 37 °C.

**Figure 3 ijms-25-04835-f003:**
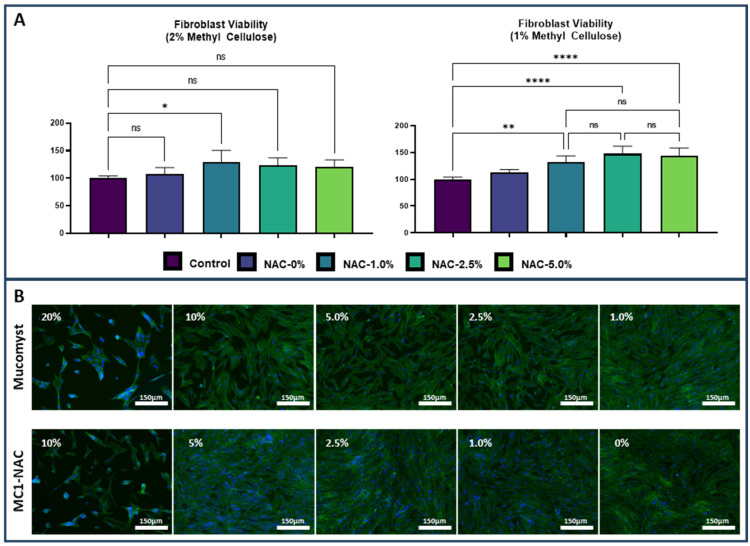
Cell viability tests using fibroblast cells show that NAC hydrogel formulations composed of 1% methylcellulose resulted in drastically improved viability compared with the NAC hydrogel formulations containing 2% methylcellulose (*n* = 5, * *p* < 0.05, ** *p* < 0.01, **** *p* < 0.0001) (**A**). This is further exemplified when observing the fluorescent-stained cells comparing cells cultured with the commercial formulation and MC1-NAC5 (scale bar = 150 µm) (**B**).

**Figure 4 ijms-25-04835-f004:**
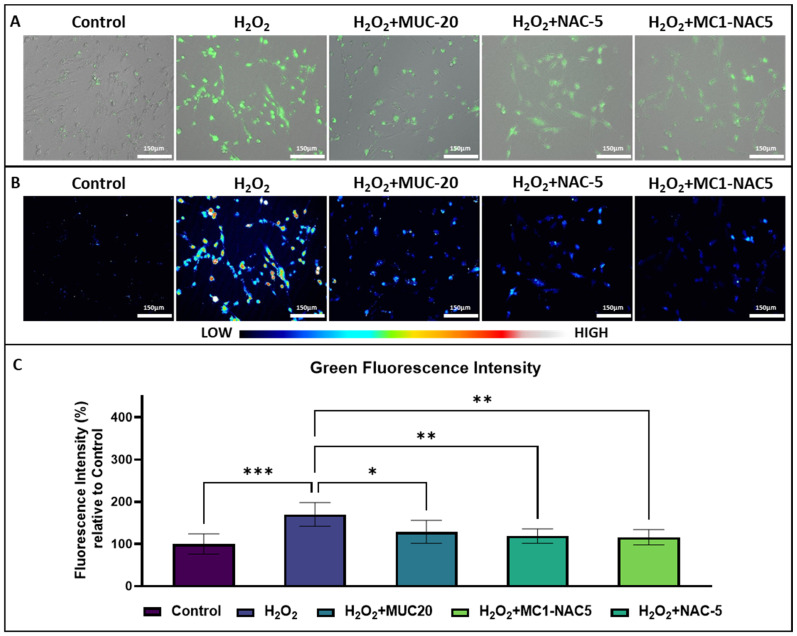
Micrographs of fibroblasts cells showing green fluorescent ROS after exposure to different treatments containing NAC (**A**). Green channels with modified LUT for improved visualization of green fluorescence from intracellular ROS (**B**) (scale bar = 150 µm). Analyses of the fluorescence intensities (**C**) show that all treatments are able to reduce ROS in H_2_O_2_-treated fibroblast cells (*n* = 6, * *p* < 0.05, ** *p* < 0.01, *** *p* < 0.001).

**Figure 5 ijms-25-04835-f005:**
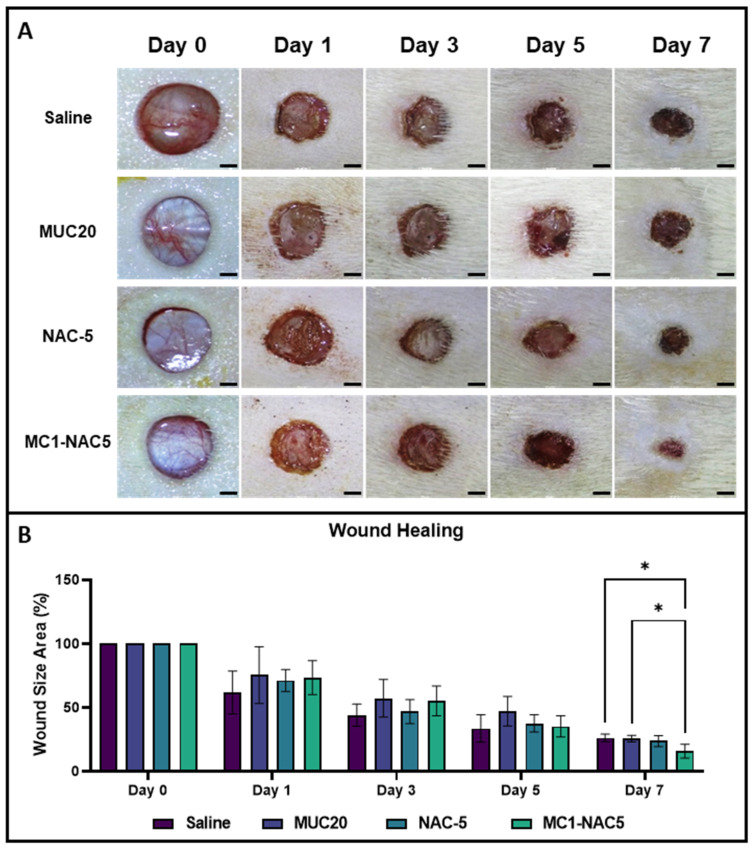
Macro images of the wound beds routinely treated with either saline, MUC20, NAC-5, or MC1-NAC5 (**A**) (scale bar = 2 mm). Results of the wound size analysis (**B**) indicate that application of the MC1-NAC5 resulted in a significantly reduced wound at day 7 but did not show any significant reduction within the 1-week observation period (*n* = 4, * *p* < 0.05).

**Figure 6 ijms-25-04835-f006:**
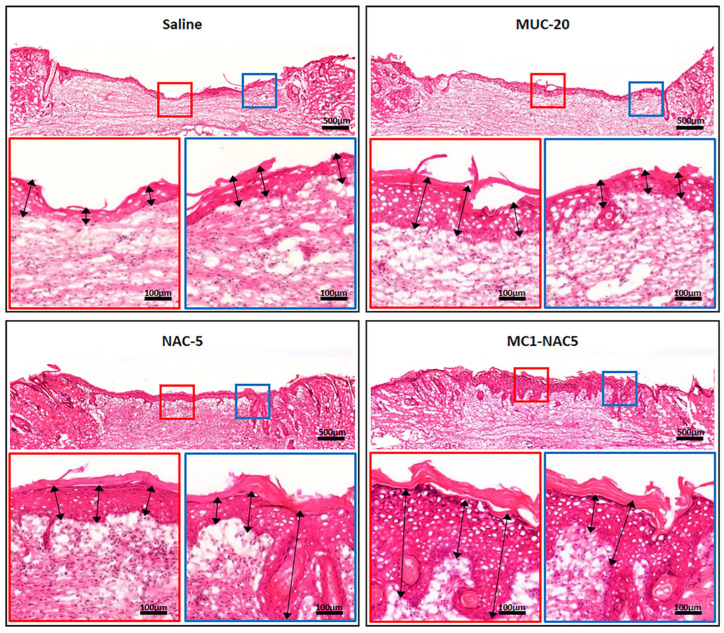
H&E-stained tissue sections of the wound bed from the different treatment groups show substantial difference in tissue development (scale bar = 500 µm). Wounds treated with the different N-acetylcysteine solutions (MUC-20/NAC5/MC1-NAC5) resulted in thicker epidermal tissue formation (black arrows). MC1-NAC5 resulted in substantially thicker epidermal tissue formation within the central wound surface (red box) and along the wound margin (blue box) (scale bar = 100 µm).

**Figure 7 ijms-25-04835-f007:**
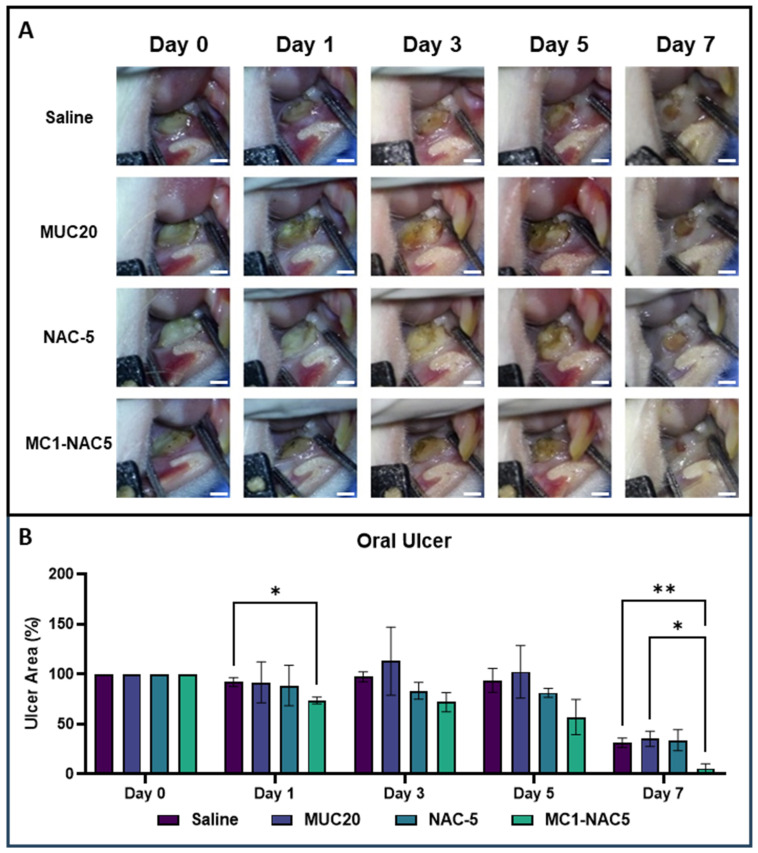
Macro images of the wound beds routinely treated with either saline, MUC20, NAC-5, or MC1-NAC5 (**A**) (scale bar = 1 mm). Results of the wound size analysis (**B**) indicate that application of the MC1-NAC5 resulted in a significantly reduced wound at day 7 but does not show any significant reduction within the 1-week observation period (*n* = 4, * *p* < 0.05, ** *p* < 0.01).

**Figure 8 ijms-25-04835-f008:**
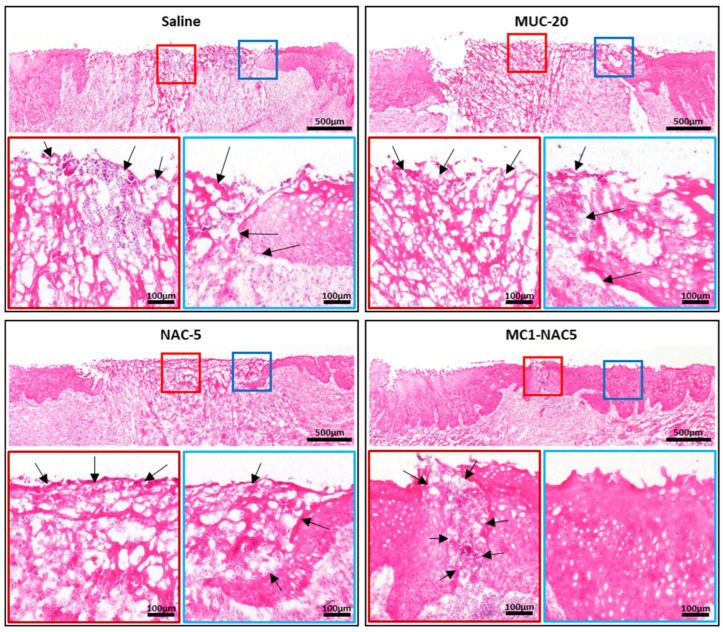
Histological analysis of the extracted oral mucosal tissue further reveals the discrepancies in tissue regenerations among different treatment regimens (scale bar = 500 µm). While all samples retained substantial fibrous tissue formation within the papillary layer (black arrows), only MC1-NAC5-treated tissue showed full closure of mucosal layer (central ulcer surface—red box; ulcer margin—blue box; scale bar = 100 µm).

**Figure 9 ijms-25-04835-f009:**
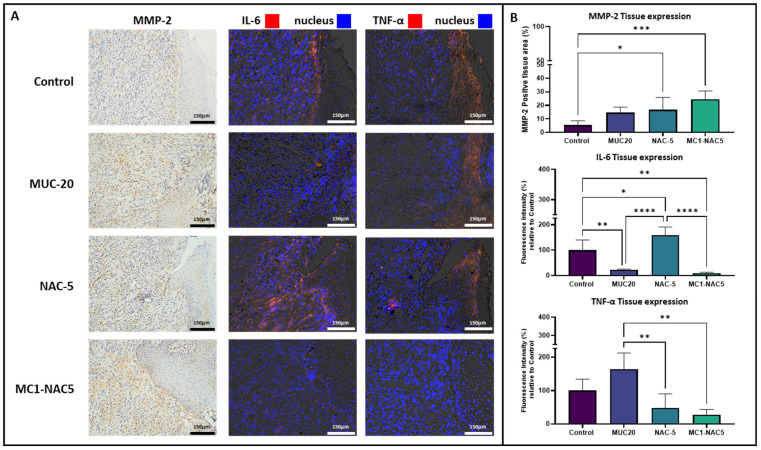
Immunostained sections of the mucosal edge of the oral ulcer from the groups show variable tissue expression among the different NAC-based treatments (**A**) (light microscope image scale bar = 150; fluorescence image scale bar = 150 µm). Image analyses based on color coverage and fluorescence indicate that oral ulcers treated with MC1-NAC5 have higher expression of MMP-2 and lowest expression of IL-6 and TNF-α (**B**) (*n* = 4, * *p* < 0.05, ** *p* < 0.01, *** *p* < 0.001 **** *p* < 0.0001).

**Table 1 ijms-25-04835-t001:** Summary of the general results gathered from this study.

Key Findings	
Effects of NAC in cells and damaged tissue is mainly dose dependent.	Application of NAC hydrogel can effectively reduce inflammation and improve tissue regeneration.
A thermo-responsive hydrogel was developed through simple combination of N-acetylcysteine and methylcellulose.	While the NAC hydrogel enhances tissue regeneration in dermal wounds, its effectiveness is even greater when used to treat oral ulcers.
Addition of methylcellulose does not compromise the antioxidant effect of NAC.	Enhancing NAC’s therapeutic impact is achievable by adjusting its viscosity and extending its retention time.

## Data Availability

Data are contained within the article.

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
