# Peer review of "Impact of Thermo-Responsive N-Acetylcysteine Hydrogel on Dermal Wound Healing and Oral Ulcer Regeneration"

_ijms, 2024, doi:10.3390/ijms25094835_

Round 1

Reviewer 1 Report

Comments and Suggestions for Authors

Authors presented the manuscript entitled Impact of Thermoresponsive N-Acetylcysteine Hydrogel on Dermal Wound Healing and Oral Ulcer Regeneration for the peer review. Wound healing is a complex, highly regulated process that is critical in maintaining the barrier function of skin. With numerous disease processes, the cascade of events involved in wound healing can be affected, resulting in chronic, non-healing wounds that subject the patient to significant discomfort and distress while draining the medical system of an enormous amount of resources.  Current study focuses on the development of a topical thermo-responsive NAC hydrogel and evaluate its efficacy for improving tissue regeneration in dermal wounds and oral  ulcers. This investigation covers the optimization of the NAC hydrogel formulation in  terms of desired gelling property to enhance retention upon application on damaged tissues and will primarily center on in vitro and in vivo biocompatibility.  In spite of very valuable experimental data, this manuscript holds several concerns .

1. Authors have neglected chronic low inflammation that prevent wound healing. At least one phrase should be added  in the Introduction.

2. One fibroblast cell line is not enough to prove cell specific effects of NAC hydrogel formulation. Please add 2 more cell lines to argue for NAC effect on them.

3. Apoptosis or comet assay should be preformed to prove cell viability after NAC aplication.

4. Moreover, I recommend to use qPCR for detailed data on genes involved in wound healing process. Authors could use tissue samples from rat oral cavities without sacrificing animals. One histological method is not enough for publication.

5. Please add method , e.g. reagent CM-H2DCFDA to assess ROS levels before and after NAC application to see the effect.

Minor comments.

All figures lack p-values after *** signs.

FIgure 3 lacks size bars in micorscopy photos.

Figure 7 needs arrows to indicate sites of changed cell surface.

4. 

Author Response

Reviewer 1

Authors presented the manuscript entitled Impact of Thermoresponsive N-Acetylcysteine Hydrogel on Dermal Wound Healing and Oral Ulcer Regeneration for the peer review. Wound healing is a complex, highly regulated process that is critical in maintaining the barrier function of skin. With numerous disease processes, the cascade of events involved in wound healing can be affected, resulting in chronic, non-healing wounds that subject the patient to significant discomfort and distress while draining the medical system of an enormous amount of resources.  Current study focuses on the development of a topical thermo-responsive NAC hydrogel and evaluate its efficacy for improving tissue regeneration in dermal wounds and oral  ulcers. This investigation covers the optimization of the NAC hydrogel formulation in  terms of desired gelling property to enhance retention upon application on damaged tissues and will primarily center on in vitro and in vivo biocompatibility.  In spite of very valuable experimental data, this manuscript holds several concerns.

  1. Authors have neglected chronic low inflammation that prevent wound healing. At least one phrase should be added in the Introduction.

Response: Thank you for your comment. We have added a statement with relevant citations pertaining to the persistence of low-level inflammation and oxidative stress in the introduction section.

“Additionally, continuous oxidative stress is linked to alterations in proteins and lipid peroxidation, which has been demonstrated to contribute to sustained low-level inflammation hindering the natural progression of wound healing and leading to elevated cell apoptosis and senescence [4-6].”

  1. One fibroblast cell line is not enough to prove cell-specific effects of NAC hydrogel formulation. Please add 2 more cell lines to argue for NAC effect on them.

Response: We acknowledge your concern. Numerous studies have already been conducted regarding the biocompatibility of NAC. We have selected fibroblast cells considering these types of cells contribute to the majority of the initial tissue formation in wounds and oral ulcers. Hence, we believe the invitro experiments involving fibroblast cells alone are enough for the initial experiments before the in vivo tests.

  1. Apoptosis or comet assay should be preformed to prove cell viability after NAC aplication.

Response: Thank you for your suggestion. We have chosen the CCK assay since is a widely adopted method for determining basic mammalian cell biocompatibility. While comet assay would be useful for observing DNA damage/ genotoxicity, it would be less relevant as a test considering CCK assay directly relates to viable cells after treatment compared to comet assay which mainly detects single-cell DNA strand breaks.

  1. Moreover, I recommend using qPCR for detailed data on genes involved in the wound healing process. Authors could use tissue samples from rat oral cavities without sacrificing animals. One histological method is not enough for publication.

Response: Thank you for your suggestion. We have provided additional histological analysis data examining the expression of 3 immunostained markers (MMP2, TNF-a, and IL-6). Please see Figure 9. Furthermore, we included an additional section presenting and discussing these results in the ‘Results and Discussion’ section, please see section X.X.

3.6. Effect of NAC-Hydrogel on Oral Ulcer Inflammation

Considering that the NAC-hydrogel showed a pronounced effect as an oral ulcer treatment, in addition to observing the resulting tissue microarchitecture from the different treatment groups, tissue expression of pro-inflammatory cytokines and tissue remodeling were also examined through immunostaining. MMP-2 was used as a maker for active tissue remodeling [37-39] while IL-6 and TNF-α were observed as pro-inflammatory markers [40-45]. IHC staining revealed elevated expression of MMP-2 along the fibrous tissue formation in all treatment groups particularly in both tissues treated with NAC-5 and MC1-NAC5. However, neither was significantly higher than the other. Although oral ulcer tissues treated with the commercial NAC solution showed higher expression of MMP-2, it was not significantly different from that of the control group. Examination of IL-6 and TNF-α through immunofluorescence staining showed drastically different outcomes across all treatment groups. Oral ulcers treated with the MUC20 showed significantly lower expression of IL-6 with MC1-NAC5. NAC-5 treatment resulted in relatively higher expression of IL-6 compared to all treatment groups and the control group. Examination of TNF-α revealed a reversed situation between MUC20 and NAC-5 groups with MUC 20 showing the highest expression of TNF-α among all treatment groups whilst not significantly differing from the control. Expressions of both aforementioned pro-inflammatory markers in MC1-NAC5 treated oral ulcers were significantly lower among all groups. These results demonstrated the importance of optimal exposure of damaged tissues to anti-oxidants to improve tissue regeneration and address chronic inflammation.”

  1. Please add method, e.g. reagent CM-H2DCFDA to assess ROS levels before and after NAC application to see the effect.

Response: We have provided additional in vitro data examining the effect of the different NAC treatments on ROS in fibroblast cells. Please see Figure 4. Furthermore, we included an additional section in the ‘Materials and Methods’ section stating how the experiment was conducted and in the ‘Results and Discussion’ section presenting and discussing these results.

2.3. ROS Fluorescence detection

Reactive Oxygen Species Detection Assay Kit (H2DCFDA) (K936-250 BioVision Inc., California, USA) was used to observe the effect of the different NAC formulation and NAC-hydrogel on the intracellular ROS in human fibroblast cells (CRL-2522, ATCC, Virginia, USA) in vitro. Fibroblast cells were seeded in a 96-well plate, with each well containing approximately 2.5x104 cells. Each group was designated with 6 wells. ROS kit staining was then performed according to the manufacturer’s protocol with relevant modifications. All cells were incubated with 100µl of 1x ROS label for 45 minutes at 37°C. The ROS label was then removed and the cells were exposed to 0.03% hydrogen peroxide (H­2O2) with or without a 10% volume of either MUC20, NAC5, or MC1-NAC5 for 1 hour. Wells treated with saline were used as a control. Micrographs of the fibroblast cells were then taken immediately after 1 hour of incubation using the EVOS 7000 imaging system (Invitrogen, Massachusetts USA). Fluorescence intensity analysis of the images was performed by separating the image channels of the micrographs and analyzing mean pixel values (n=6).”

3.3. Effect of NAC-Hydrogel on fibroblast ROS

The antioxidant effect of the NAC-hydrogel was tested against the commercial product and NAC solution at 5% concentration. Chemically induced intracellular ROS was measured in fibroblast cells exposed to hydrogen peroxide. Figure 4 A and B show the images of fibroblast cells treated with H2DCFDA after exposure to different preparations of NAC. Results indicate that all NAC treatments were relatively effective in reducing chemically induced intracellular ROS in fibroblast cells. Both NAC-5 and MC1-NAC5 treatments resulted in lower ROS-stimulated fluorescence compared to the commercial product. However, statistical analyses of the relative mean fluorescence (Figure 4C) from the sampled images revealed no significant difference among the treatment groups. These results indicate that even with the modified preparation of NAC in the form of the NAC-hydrogel, its capacity to reduce intracellular ROS is not affected and the addition of the methyl cellulose does not affect its overall biocompatibility. In addition, the lower concentration of N-acetylcysteine in the NAC-hydrogel is much more beneficial than the higher concentrations found in the commercial product.”

Minor comments.

  • All figures lack p-values after *** signs.

Response:  We have provided the respective p-value range for graph * annotations.

  • FIgure 3 lacks size bars in micorscopy photos.

Response: Scale bars have been added in all microscopy photos.

  • Figure 7 needs arrows to indicate sites of changed cell surface.

Response: We have indicated arrows to emphasize epidermal tissue differences in Figure 6 and fibrous tissue persistence in Figure 8.

Reviewer 2 Report

Comments and Suggestions for Authors

1. Please add if possible some clinical patient cases with preoperative and postoperative management with NAC. It will attract the reader.

2. What is the molecular mechanism of action?

3. What are the probable side effects and drug interactions and how to avoid that?

4. It also has an anticlotting property which causes bleeding how you can defend this?

Author Response

Reviewer 2

  1. Please add if possible some clinical patient cases with preoperative and postoperative management with NAC. It will attract the reader.

Response: Thank you for the feedback. We have briefly discussed this in the later half of the results and discussion. Currently, we consider the application of NAC-hydrogel only within the context of oral ulcer, and general wound management. Please see the highlighted sections.

  1. What is the molecular mechanism of action?

Response: Thank you for noticing this. We have included the mechanism of action in the ‘Results and Discussion” section. Please see the highlighted section.

  1. What are the probable side effects and drug interactions and how to avoid that?

Response: Thank you for your comment. We haven’t tested its interactions with other drugs, however considering extensive clinical studies have been performed using NAC in metabolically compromised patients, we believe it has minimal effect/ interactions with other medications. We have briefly discussed this as a limitation in the ‘Results and Discussion’ section. Please see the highlighted sections.

  1. It also has an anticlotting property which causes bleeding how you can defend this?

Response: We acknowledge your concern. Although NAC has been known to affect clotting mechanisms, the NAC-hydrogel that we developed has a considerably low concentration of N-acetylcysteine. As exemplified by our in vivo results, it was not able to severely compromise the formation of stable clots in fully excised skin tissues. We have briefly discussed this as a limitation in the ‘Results and Discussion’ section. Please see the highlighted sections.

Reviewer 3 Report

Comments and Suggestions for Authors

To: International Journal of Molecular Sciences (IJMS)

Dear EIC,

And dear AE,

This is my review results for the manuscript ID: ijms-2969803.

This is an In vivo study to evaluate a “Thermoresponsive N-Acetylcysteine Hydrogel on 2

Dermal Wound Healing and Oral Ulcer Regeneration”. These techniques were applied to reach the aims of the study: cell culture and preparation, cell viability assay, H&E histological staining, and related statistical analysis using the GraphPad Prism tool. The results are interesting and I think they can add some novel things to the field. The results were discussed adequately and concluded more useful. I read the manuscript several times revised it carefully and prepared some minor and major comments.

Comments

Minor comments

·      Suggested title “In-vivo impact of Thermoresponsive N-Acetylcysteine Hydrogel on 2 Dermal Wound Healing and Oral Ulcer Regeneration”

·      Kindly check line 152: I think this should have a separate subsection.

·      Kindly check line 168: I think this should have a separate subsection.

·      Figures were addressed by “Fig.X” in the manuscript body; however, Figure 3 did not follow this routine. Kindly correct this in lines 303-304.

Major comments

·      For easier access to the results of the work I strongly recommend abstracting the findings in a separate table.

·      I didn’t see any keywords. Kindly check that.

·      Why the authors didn’t describe the chemistry of the therapeutics that were used in this study?

·      Are any further methods needed to apply in this study to describe the chemical aspects of therapeutics? Such as spectrophotometry?

Good luck.

Author Response

Reviewer 3

This is an In vivo study to evaluate a “Thermoresponsive N-Acetylcysteine Hydrogel on 2 Dermal Wound Healing and Oral Ulcer Regeneration”. These techniques were applied to reach the aims of the study: cell culture and preparation, cell viability assay, H&E histological staining, and related statistical analysis using the GraphPad Prism tool. The results are interesting and I think they can add some novel things to the field. The results were discussed adequately and concluded more useful. I read the manuscript several times revised it carefully and prepared some minor and major comments.

Comments

Minor comments   

Proposed Title: "In-vivo Impact of Thermoresponsive N-Acetylcysteine Hydrogel on Dermal Wound Healing and Oral Ulcer Regeneration"
Response: We appreciate your suggestion. While the proposed title adds specificity, we believe the current title offers broader appeal.

Regarding Line 152: "I think this should have a separate subsection."
Response: Thank you for your observation. We have addressed this concern by segregating the animal experiments for dermal wounds and oral ulcers. These have been placed under distinct subsections, namely '2.4.1 Induction and Treatment of Dermal Wounds' and '2.4.2 Induction and Treatment of Oral Ulcers', respectively, within the 'Materials and Methods' section.
Thank you once again for your valuable feedback.

  • Kindly check line 168: I think this should have a separate subsection.

Response: Thank you for noticing this. We have organized the section numbering across within the manuscript to comply with the current format of the journal. Please see the highlighted subsections.

  • Figures were addressed by “Fig.X” in the manuscript body; however, Figure 3 did not follow this routine. Kindly correct this in lines 303-304.

Response: We appreciate this comment. We have changed the ‘Fig.X’ format within the body text to ‘Figure X’ in accordance with the current format of the journal. Please see the highlighted sections.

Major comments

  • For easier access to the results of the work I strongly recommend abstracting the findings in a separate table.

Response: We acknowledge your concern. We have summarized our key findings in Table 1.

  • I didn’t see any keywords. Kindly check that.

Response: Thank you for noticing. We have added keywords within the manuscript. Please see the highlighted section.

“hydrogel, n-acetylcysteine, oral ulcer, wound healing, inflammation”

  • Why the authors didn’t describe the chemistry of the therapeutics that were used in this study?

Response: We acknowledge your concern. We have added the mechanism of the therapeutics in the latter half of the ‘Results and Discussion’ section. Please see the highlighted sections.

  • Are any further methods needed to apply in this study to describe the chemical aspects of therapeutics? Such as spectrophotometry?

Response: Thank you for your comment. The methyl cellulose component of the NAC-hydrogel generally serves as a viscosity/ physical form modifier and has minimal effect on the biological activity of the NAC component. This is explicitly demonstrated in the experiment for observing the reduction of intracellular ROS in vitro. Hence, there is no further need for extensive characterization of the final tested sample. Additional staments pertaining to this have been added in the ‘Results and Discussion’ Section.

Good luck.

Round 2

Reviewer 1 Report

Comments and Suggestions for Authors

Authors have addressed all my comments. Thank you.

Reviewer 3 Report

Comments and Suggestions for Authors

The author's responses convinced me and I have no further questions or comments.

Good luck